# Interrogating the precancerous evolution of pathway dysfunction in lung squamous cell carcinoma using XTABLE

**Matthew Roberts[1,2], Julia Ogden[2,3], AS Mukarram Hossain[1,2], Anshuman Chaturvedi[2,4], Alastair RW Kerr[1,2], Caroline Dive[1,2], Jennifer Ellen Beane[5], Carlos Lopez-Garcia[2,3]***

[1]Cancer Biomarker Centre, Cancer Research UK Manchester Institute, The University of Manchester, Macclesfield, United Kingdom; [2]Cancer Research UK Lung Cancer Centre of Excellence, Alderley Park, United Kingdom; [3]Translational Lung Cancer Biology Laboratory, Cancer Research UK Manchester Institute, University of Manchester, Macclesfield, United Kingdom; [4]Department of Histopathology, The Christie Hospital, Manchester, United Kingdom; [5]Boston University School of Medicine, Boston, United States

**Abstract** Lung squamous cell carcinoma (LUSC) is a type of lung cancer with a dismal prognosis that lacks adequate therapies and actionable targets. This disease is characterized by a sequence of low- and high-grade preinvasive stages with increasing probability of malignant progression. Increasing our knowledge about the biology of these premalignant lesions (PMLs) is necessary to design new methods of early detection and prevention, and to identify the molecular processes that are key for malignant progression. To facilitate this research, we have designed XTABLE (E**x**ploring **Tr**anscriptomes of **B**ronchial **L**esions), an open-source application that integrates the most extensive transcriptomic databases of PMLs published so far. With this tool, users can stratify samples using multiple parameters and interrogate PML biology in multiple manners, such as two- and multiple-group comparisons, interrogation of genes of interests, and transcriptional signatures. Using XTABLE, we have carried out a comparative study of the potential role of chromosomal instability scores as biomarkers of PML progression and mapped the onset of the most relevant LUSC pathways to the sequence of LUSC developmental stages. XTABLE will critically facilitate new research for the identification of early detection biomarkers and acquire a better understanding of the LUSC precancerous stages.

***For correspondence:**
carlos.lopezgarcia@cruk.
manchester.ac.uk

**Competing interest:** The authors declare that no competing interests exist.

## Editor's evaluation

The authors have created a resource tool that is valuable in assessing precancerous lesions in the lung, which may serve as a tool for investigators working in this area, and as an example for additional similar resources. The accessibility of the tool is a concern but does not diminish the quality of the product.

## Introduction

Lung squamous cell carcinoma (LUSC) is a type of non-small cell lung cancer that accounts for 20–30% of all lung cancer cases (*Chen et al., 2014*; *Sung et al., 2021*; *Bray et al., 2021*). Despite being the second most frequent type of lung cancer (*Torre et al., 2016*), our knowledge regarding the biology

**eLife digest** Lung squamous cell carcinoma is the second most common lung cancer. However, very little is known about how normal tissues in the lung develop in to these tumours. Like many cancers, this transformation comprises of an intermediate phase where healthy cells begin to form lesions that may (or may not) progress in to tumours. Understanding the biology of these lesions in lung squamous cell carcinoma may help clinicians detect them before they become cancerous.

Knowing which genes are switched on and off during this intermediary phase can provide clues as to how these lesions form. There are already some publicly available transcriptional datasets showing the activity of tens of thousands of genes in pre-cancerous lesions extracted from patients with lung squamous cell carcinoma. But not every laboratory has the bioinformatic tools and skills required to interrogate these extensive databases.

To address this, Roberts et al. built an open-source platform called XTABLE (short for Exploring Transcriptomes of Bronchial Lesions) which can analyse transcriptional datasets in multiple ways depending on the needs of the user. For instance, the tool can stratify the data into groups based on different parameters, such as the lesions potential to progress in to cancer, to see how the genes of the groups compare. It can also analyse the activity of individual genes and sets of genes involved in the same biological processes.

Using XTABLE, Roberts et al. showed that a biological process linked to lung squamous cell carcinoma is also involved in the formation of pre-cancerous lesions. This suggests that molecules and genes associated with this process could potentially help scientists design prevention strategies.

XTABLE will help researchers to better understand the biology of pre-cancerous lesions and how they develop in to tumours. Moreover, it will make it easier for scientists to validate their hypotheses using data collected from patients. The tool could also be useful for scientists interested in other types of lung cancers that share a similar biology.

of this disease as well as the therapeutic modalities to treat it remain far behind the most frequent type of lung cancer, lung adenocarcinoma (LUAD) (*Hirsch et al., 2016*; *Khuder, 2001*).

LUAD genetics is dominated by mutations (that are often druggable) that activate the RTK/RAS pathway, including EGFR and KRAS mutations (*The Cancer Genome Atlas Research Network, 2012*; *Jamal-Hanjani et al., 2017*). However, the genetic landscape of LUSC is more complex, with multiple pathways altered in subsets of patients and a lack of actionable mutations (*Campbell et al., 2016*; *The Cancer Genome Atlas Research Network, 2012*; *Kim et al., 2014*), precluding the development of new therapies. Hence, the only pharmacological therapies available to treat LUSC patients are immune checkpoint inhibitor monotherapy or in combination with chemotherapy (*Mok et al., 2019*; *Paz-Ares et al., 2018*; *Weinberg and Gadgeel, 2019*). Furthermore, the National Lung Cancer Matrix Trial and The Lung Master Protocol, the largest personalized medicine trials in lung cancer, have not shown clear therapeutic benefits with targeted agents in LUSC (; *Middleton et al., 2020*; *Redman et al., 2020*). LUSC also has a worse prognosis than LUAD independent of stage at detection, with a 5-year overall survival (OS) of 6.2% for patients diagnosed with distant metastasis (9.5% for LUAD). However, patients diagnosed with localized disease are eligible for curative surgery and the 5-year OS is 50% (*National Cancer Institute and DCCPS, 2018*). Therefore, early detection is currently the most valuable tool to prevent deaths by LUSC as evidenced by several ongoing programmes of lung cancer early detection (*Crosbie et al., 2019a*; *Crosbie et al., 2019b*; *Aberle et al., 2011*). These initiatives make use of low-dose CT scans in high-risk populations, but in spite of the frequent detection of localized lung cancer eligible for resection with curative intent, 40% of early diagnosed patients die within 5 years.

LUSC progresses through a series of premalignant stages characterized by alterations of the normal bronchial epithelium (*Figure 1*; *Ishizumi et al., 2010*; *Kadara et al., 2016*; *Thakrar et al., 2017*; *Pennycuick et al., 2020*). These endobronchial premalignant lesions (PMLs) are classified as low-grade (squamous metaplasia, mild and moderate dysplasia) and high-grade (severe dysplasia, and carcinomas in situ [CIS]) (*Figure 1*). However, not all PMLs progress to LUSC. Although obvious differences exist between the multiple studies published on the topic, high-grade PMLs have a higher risk of progression than low grade, and high levels of chromosomal instability (CIN) are also predictive

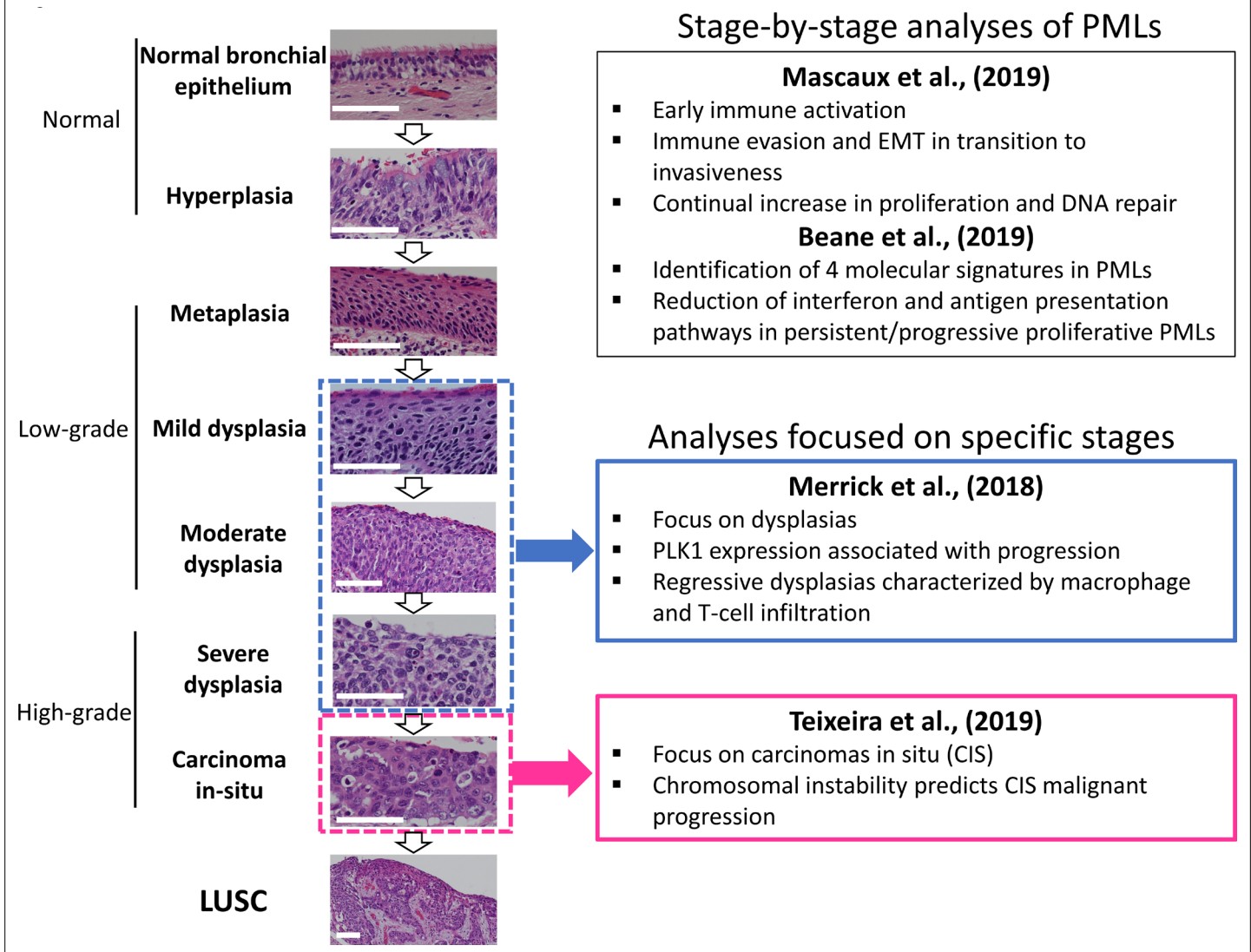

**Figure 1.** Developmental stages of lung squamous cell carcinoma (LUSC) premalignant lesions (PMLs) with representative histological images for each stage (haematoxilyn-eosin) and a summary of the four studies included in XTABLE (Exploring Transcriptomes of Bronchial Lesions). PMLs are typically classified as normal epithelium (including hyperplasia), low-grade and high-grade. Two studies (*Mascaux et al., 2019*, and *Beane et al., 2019*) carried out gene expression analysis of multiple developmental stages, whereas *Merrick et al., 2018*, and *Teixeira et al., 2019*, focused on dysplasias (blue boxes) and carcinomas in situ (CIS) (pink boxes), respectively. The most relevant findings of each article are summarized in the figure. Error bars=50 μm.

of progressive PMLs (*Thakrar et al., 2017*; *van Boerdonk et al., 2014*; *Teixeira et al., 2019*; *Merrick et al., 2016*). This potential role of CIN as biomarker of PMLs progression has been observed in low- and high-grade PMLs (*van Boerdonk et al., 2014*; *Teixeira et al., 2019*). These reports showed that high levels of copy number variations was the best predictor of progressive PMLs (*van Boerdonk et al., 2014*; *Teixeira et al., 2019*) and that immune response is the most likely cause of regression as these lesions contained higher levels of immune infiltration (*Pennycuick et al., 2020*; *Beane et al., 2019*). These results are supported by transcriptomic analysis of PMLs that indicate immune evasion in the transition to invasiveness (*Mascaux et al., 2019*).

The detection of PMLs cannot be carried out by routine patient imaging techniques such as CT or PET scans as the morphological change that they cause in the airway does not result in radiological contrast. Alternatively, ablation of high-grade PMLs detected by autofluorescence bronchoscopy using minimally invasive endoscopic procedures in high-risk populations is an innovative and interesting strategy to prevent LUSC (*Guibert et al., 2016*). Nonetheless autofluorescence bronchoscopy is an expensive and complex technique of limited use in large screening programmes. Therefore, simpler,

more cost effective, and scalable methods of high-grade PML detection are needed to prevent deaths by LUSC. Improving the detection of PMLs requires a better understanding of their biology and the validation of adequate biomarkers of progressive lesions that can be translated into new technologies for large screening initiatives. Cell surface proteins, metabolites, nasal-based biomarkers, blood and sputum/bronchoalveolar lavage biomarkers are examples of biomarkers that can be used to improve and/or complement current diagnostic techniques (such as CT and PET scans) or develop new ones. A user-friendly application to interrogate gene expression from studies focusing on precancerous LUSC stages will enhance the advances in PML biology. However, such application does not exist.

Recently, scientific interest in the biology of preinvasive LUSC stages has motivated the publication of several articles characterizing PML transcriptomes from various perspectives (*Figure 1*). Two reports published by *Beane et al., 2019*, and *Mascaux et al., 2019*, showed stage-by-stage gene expression analyses of PMLs. These studies provided the most detailed transcriptomic characterization of all PML stages so far and identified changes in the immune microenvironment associated with invasive transformation. Additionally, longitudinal studies by *Beane et al., 2019*, *Merrick et al., 2018*, and *Teixeira et al., 2019*, contained samples with known progression potential. *Beane et al., 2019*, identified molecular subtypes with specific biological traits whereas articles published by *Merrick et al., 2018*, and *Teixeira et al., 2019* focused on gene expression of specific preinvasive stages (dysplasias and CIS, respectively) with the objective of identifying predictors of PML progression and the biomolecular processes involved (*Figure 1*). These studies provide a valuable source of gene expression data to identify candidate biomarkers for the detection of high-risk PMLs and/or early stage LUSC as well as to investigate the biology of premalignant LUSC progression.

Transparent and straightforward accessibility to transcriptomic databases is a key requirement for the open science philosophy. Applications that integrate multiple databases focusing on the same biological and clinical problem, such as camcAPP (*Dunning et al., 2017*), allow cross-comparisons between independent studies, strengthen the robustness of results obtained, and allow the selection of high-confidence data. In this report, we provide an overall description of XTABLE (E**x**ploring **Tra**nscriptomes of **B**ronchial **L**esions) and provide examples of its functions. XTABLE is a new open-source application that will enable scientists to interrogate currently four LUSC PML transcriptomic datasets in a versatile manner that can be adapted to the needs of each researcher. Specifically, LUSC prevention and diagnosis are the main areas that can benefit the most from XTABLE, but its versatility and multiple functions lend themselves to the exploration of a variety of research questions. Without XTABLE, researchers would have to put together all the packages, data processing steps themselves for each analysis they wished to run. In this report, we provide an overall explanation of all the functions of XTABLE as well as a detailed description of the most important analysis modules, including two-group comparisons, gene-of-interest analyses, and interrogation of transcriptional signatures. Additionally, we explored the use of CIN-related signatures as a biomarker of progressive PMLs and mapped the onset of the most important LUSC pathways to its developmental stages.

**Table 1.** Description of the four cohorts included in XTABLE (E**x**ploring **Tra**nscriptomes of **B**ronchial **L**esions).

| GEO accession | PMID | Stages | Progression status known | Number of samples | Sample type | Transcriptome |
|---|---|---|---|---|---|---|
| GSE33479 | 31243362 | Multiple | No | 122 | Whole biopsies | Microarray |
| GSE109743 | 31015447 | Multiple* | Yes | 448 Discovery cohort (197 Bx, 91 Br) Validation cohort (111Bx, 49 Br) | Whole biopsies and brushings | RNAseq |
| GSE114489 | 29997230 | Dysplasias normal | Yes | 63 | Whole biopsies | Microarray |
| GSE108124 | 30664780 | CIS | Yes | 33 | Microdissected | Microarray |

Bx: biopsies; Br: brushings.
*This cohort includes neither carcinomas in situ (CIS) nor invasive carcinomas.

## Results

### Description of the studies included in XTABLE

Four datasets originating from four independent PML transcriptomic studies have been included in the XTABLE application (*Figure 1*, *Table 1*). Two datasets, GSE109743 (*Beane et al., 2019*) and GSE33479 (*Mascaux et al., 2019*), provide gene expression data of the developmental LUSC stages (with and without progression status information, respectively), whereas the remaining studies, GSE114489 (*Merrick et al., 2018*) and GSE108124 (*Teixeira et al., 2019*), focus on analysing specific PML stages (dysplasias and CIS, respectively) that have been followed up to establish their progressive, persistent, or regressive potential.

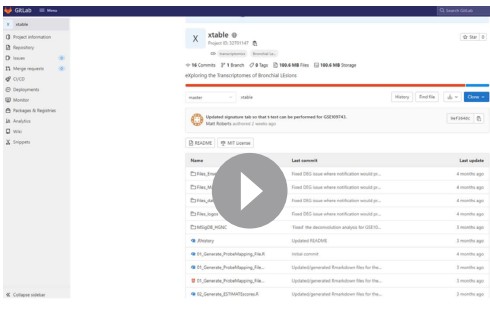

1. Go to **https://gitlab.com/cruk-mi/xtable**

**Video 1.** Step-by-step instructions to install XTABLE (Exploring Transcriptomes of Bronchial Lesions) using RStudio.
https://elifesciences.org/articles/77507/figures#video1

Dataset GSE33479 (*Mascaux et al., 2019*) comprises expression microarray analyses of 122 endobronchial biopsies of unknown progression status. Samples were obtained following autofluorescence bronchoscopy and as no enrichment or microdissection of the biopsy epithelium was carried out, variable levels of stromal component are present in samples. Although this results in dilution of epithelial signals, it has the advantage of providing information about the microenvironment as well as insights into the infiltrated immune cells. Dataset GSE109743 (*Beane et al., 2019*) consists of a transcriptomic analysis of whole PML biopsies with no purification of the epithelial compartment as well as bronchial brushings obtained from adjacent normal regions of the bronchial mucosa. Unlike GSE33479, GSE109743 used RNAseq and more importantly, includes the progression status for some lesions established by serial biopsies. Samples were classified as 'normal-stable' when they changed between normal, hyperplasia, and metaplasia, 'regressive' when they regress from dysplasia to a less severe dysplastic grade, or from dysplasia to normal/hyperplasia/metaplasia. Remaining samples were considered persistent/progressive. One advantage of this dataset is the large number of samples divided into a discovery and a validation cohort (*Figure 1*, *Table 1*). However, the representation of each PML stage is not homogeneous and CIS samples are not included.

Datasets GSE114489 (*Merrick et al., 2018*) and GSE108124 (*Teixeira et al., 2019*) constitute a different type of study. Both make use of sequential biopsies to classify lesions according to their progression potential, but they differ in the stages included, the classification of progression status and sample processing. In GSE114489, the authors collected 63 baseline bronchial biopsies (with corresponding follow-up biopsies) and classified samples in four groups: 23 persistent dysplasias (dysplastic lesions with the same or higher severity scores in follow-up biopsies), 15 regressive dysplasias (dysplasias progressing to lower severity scores), 9 progressive non-dysplasias (biopsies with normal or hyperplastic morphologies that progress to more severe morphologies), and 16 stable non-dysplasias (normal or hyperplastic pathology that remains stable in follow-up biopsies). Microarray analysis of gene expression was performed on whole biopsies. The study published by *Teixeira et al., 2019* (GSE108124) also makes use of follow-up biopsies to classify the progression potential of PMLs, but unlike GSE114489, GSE108124 focuses on CIS and progression is defined as the transition to invasive carcinomas in follow-up biopsies. RNA for gene expression microarray analysis was extracted from microdissected FFPE samples to enrich the epithelial component.

### XTABLE access, interface, and functions

XTABLE download can be carried out from https://gitlab.com/cruk-mi/xtable (copy archived at *Roberts, 2022*) following the instructions in the Materials and methods section and *Video 1*. XTABLE has been designed using the shiny app interface (see Materials and methods section) and its functions have been divided into 11 interrelated tabs that contain a specific function to interrogate each dataset separately (*Figure 2A*).

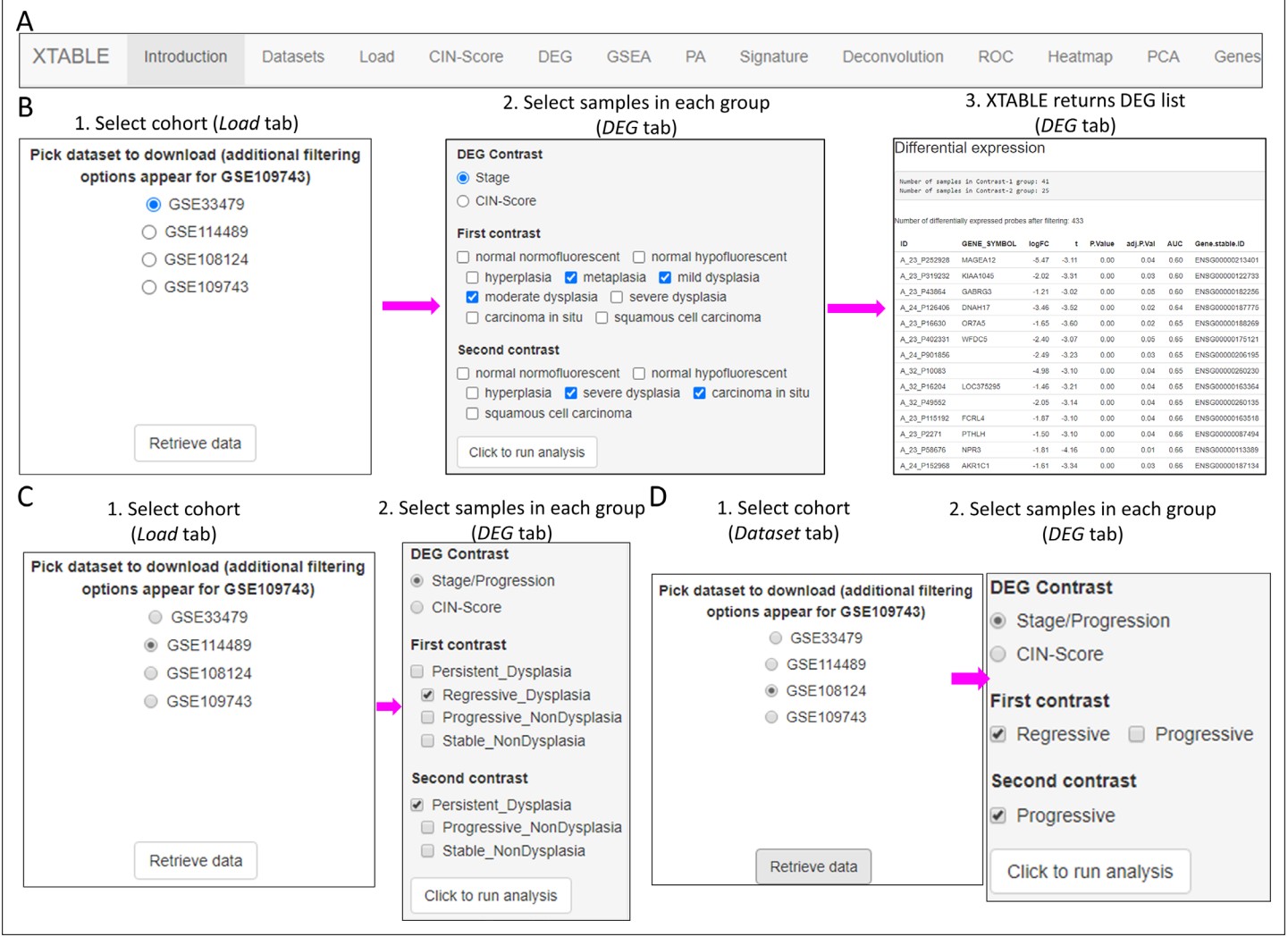

**Figure 2.** Overall organization of XTABLE (**E**xploring **T**ranscriptomes of **B**ronchial **L**esions) functions and use of the *DEG* function. (**A**) Organization of all the functions in the XTABLE interface. The functions are interrelated and completing certain analyses requires the use of several functions. For instance, the *GSEA* and *PA* functions operate with gene lists obtained with the *DEG* function. (**B**) Workflow to obtain differentially expressed genes between two groups using the *DEG* function. The example shows groups of samples arranged by developmental stage to compare low-grade and high-grade premalignant lesions (PMLs) in the GSE33479 cohort. (**C and D**) Workflow to obtain differentially expressed genes between two groups using the *DEG* function. The two groups have been arranged by progression status using in the GSE114489 and GSE108124 cohorts, respectively.

The online version of this article includes the following figure supplement(s) for figure 2:

**Figure supplement 1.** Sample selection options for cohort GSE109743.

**Figure supplement 2.** Visualization of chromosomal instability (CIN)-scores in with XTABLE (**E**xploring **T**ranscriptomes of **B**ronchial **L**esions).

**Figure supplement 3.** Example of receiver operating characteristic (ROC) curves visualization for a gene of interest (NRTK2) in premalignant lesion (PML) samples stratified by low and high grades.

## Introduction

Detailed description of the four studies, analyses performed in each tab, output formats, links to the four articles used in the application, bioinformatic packages used in the different functions and additional references.

## Dataset tab

Explanation of the differences in methodology and progression status definitions between the four studies.

## Load tab

In this tab, the user selects the database (GSE114489, GSE108124, GSE109743, or GSE33479) for subsequent interrogation. Due to specific differences between the four studies, only one dataset can be interrogated at a time. Since dataset GSE109743 contains a discovery and a validation cohort, as well as biopsies and brushings, preselection of the cohort and type of sample has to be carried out (*Figure 2—figure supplement 1*).

## CIN-score tab

CIN has been identified as a good predictor of PML progression. Except for *Teixeira et al., 2019*, the other three studies do not provide genomic analyses that provide an estimate of the level of chromosomal alterations in the samples. Several gene expression signatures that correlate with CIN (CIN-scores) have been described in the literature (*Teixeira et al., 2019*; *Carter et al., 2006*). This *CIN-score* function returns a list of several CIN-scores (CIN70, CIN25, and CIN5, depending on the number of genes included in the signature) for all samples included in the study and a graph depicting CIN-scores in different sample types classified according to the sample classification defined in the study. A line marking a selected CIN-score threshold can be added for visualization purpose and to help determine appropriate thresholds for additional CIN-related analyses (*Figure 2—figure supplement 2*).

## *DEG* tab

Function to identify differentially expressed genes in comparisons of two groups of samples determined by the user. This function allows the selection of p-value and fold-change cutoffs for the analysis. Additionally, AUC is calculated for each gene to provide further confidence to the differentially expressed gene results and additional gene IDs are added at this stage that have been sourced from two separate datasets to maximize identification and discrepancies manually reviewed to improve downstream enrichment and pathway analyses.

## GSEA and PA

In these two tabs, the user can carry out gene set and pathway enrichment analyses using multiple tools (goseq/ideal, fgsea/MSigDB, enrichR, gage/gageData, kegga/pathview, ReactomePA, Progeny, and Dorothea). This function operates with the list of differentially expressed genes obtained with the *DEG* function.

## Signature

This function returns the gene expression values of a user-defined gene list. Lists can be manually entered or uploaded from a .csv file.

## Deconvolution

Estimation of immune and stromal component in samples from gene expression data using the ESTI-MATE tool.

## ROC

This function returns receiver operating characteristic (ROC) curves in a user-defined comparison of two groups of samples stratified by the expression of a gene or by CIN-scores.

## Heatmap

Returns a gene expression heatmap for genes selected by variance, user-defined gene signatures, and differentially expressed genes in the *DEG* tab.

## PCA

Principal component analysis (PCA) of all the samples included in each study. Several sample characteristics can be highlighted in the PCA plot including CIN signatures, progression status, and PML stage.

## Gene

Statistical analysis of expression of individual genes selected by the user. Several options for sample groupings are available.

### Two-group differential expression analysis

The discovery of candidate biomarkers for the detection of PMLs at high risk of malignant progression and the interrogation of PML biology depends greatly on the comparison of gene expression profiles between lesions with known progressive or regressive potential. The four databases included in XTABLE contain different types of information which influence how users can interrogate these databases. To facilitate the interrogation of the four transcriptomic databases in a manner that allows the versatile stratification of samples, we have designed a module in XTABLE named *DEG* (*Figure 2B–D*), that returns the differentially expressed genes in two user-defined groups of samples stratified by PML stage (GSE33479 and GSE109743), by known progression status (GSE109743, GSE114489, and GSE108124) or by CIN-score thresholds. In XTABLE, we have included three CIN-scores, named CIN70, CIN25 (*Carter et al., 2006*), and CIN5 (*Teixeira et al., 2019*). CIN70 and CIN25 have been reported in the literature, with the former containing the greatest number of genes for interrogation. CIN5 is derived from the signature used by *Teixeira et al., 2019*, and is reported to show a good correlation with progression in CIS lesions.

Setting contrast groups by stage using *DEG* allows the comparison of two individual PML stages or the grouping of multiples stages into two groups. For instance, to compare the differential expression between low-grade and high-grade PMLs, we can define two contrast groups, one including metaplasia, low and moderate dysplasia (low-grade) and one including severe dysplasia and CIS (high-grade) using cohort GSE33479 (*Figure 2B*). After selection of the cohort and setting up the groups (*Figure 2B*), the application returns a downloadable list of differentially expressed genes with associated statistical information (*Figure 2B*), including AUC inferred by ROC analysis. ROC curves associated with each gene in a two-group comparison, useful for biomarker discovery, can be downloaded using the *ROC* tab (*Figure 2—figure supplement 3*). Straightforward grouping of progressive and regressive samples can also be carried out by selecting the 'Progression' contrast in studies that provide this information (*Figure 2C and D*). To carry out two-group comparison by CIN signatures, the *CIN-score* tab provides a graphic visualization of the three CIN-scores for all samples to guide the selection of a CIN-score threshold for sample stratification (*Figure 3A*). A .csv file containing the CIN-scores of all samples can be downloaded. The selected CIN threshold can be used in the *DEG* tab to stratify CIN-high and low samples and retrieve a list of differentially expressed genes (*Figure 3A*).

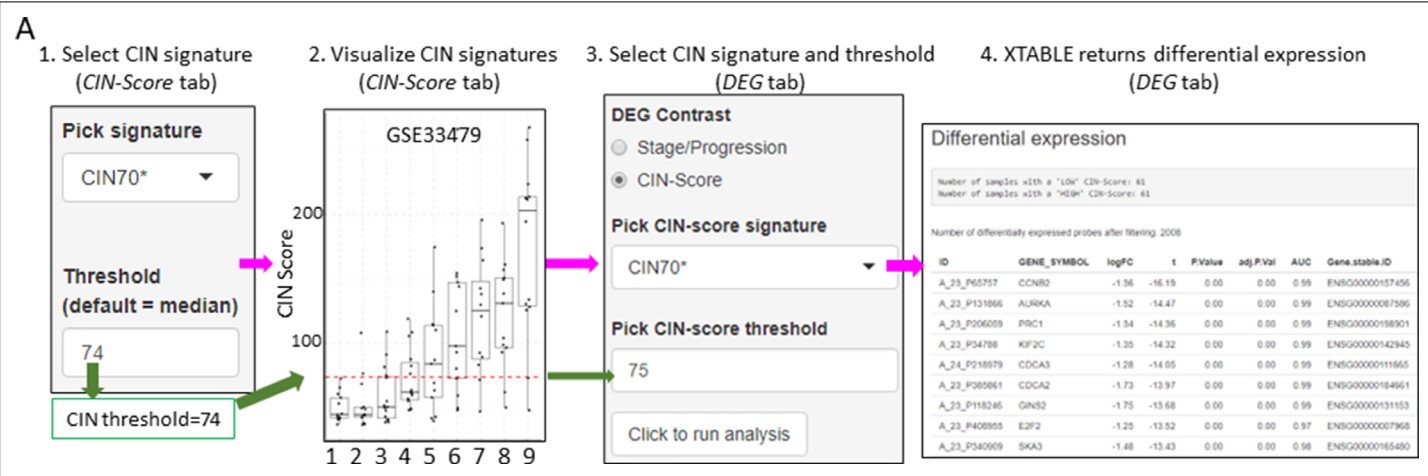

**Figure 3.** Differential expression analysis between two groups of samples classified according to a chromosomal instability (CIN)-score threshold. The *CIN-score* function allows the graphic visualization of CIN-scores for all samples in a study. A CIN-score threshold selected by the user can be depicted on the graph (red dotted line). The CIN-score threshold selected by the user can be used in the *DEG* tab to define the two-group comparison. Stages 1–9 represent the nine developmental stages of LUSC as described in *Mascaux et al., 2019* (GSE33479). CIN70, CIN25, and CIN5 can be used in the *DEG* tab. Sample sizes: n=12 (stage 7), n=13 (stages 1, 5, 6 and 8), n=14 (stages 2 and 9), n=15 (stages 3 and 4). Boxplots show median and upper/lower quartile. Whiskers show the smallest and largest observations within 1.5× IQR.

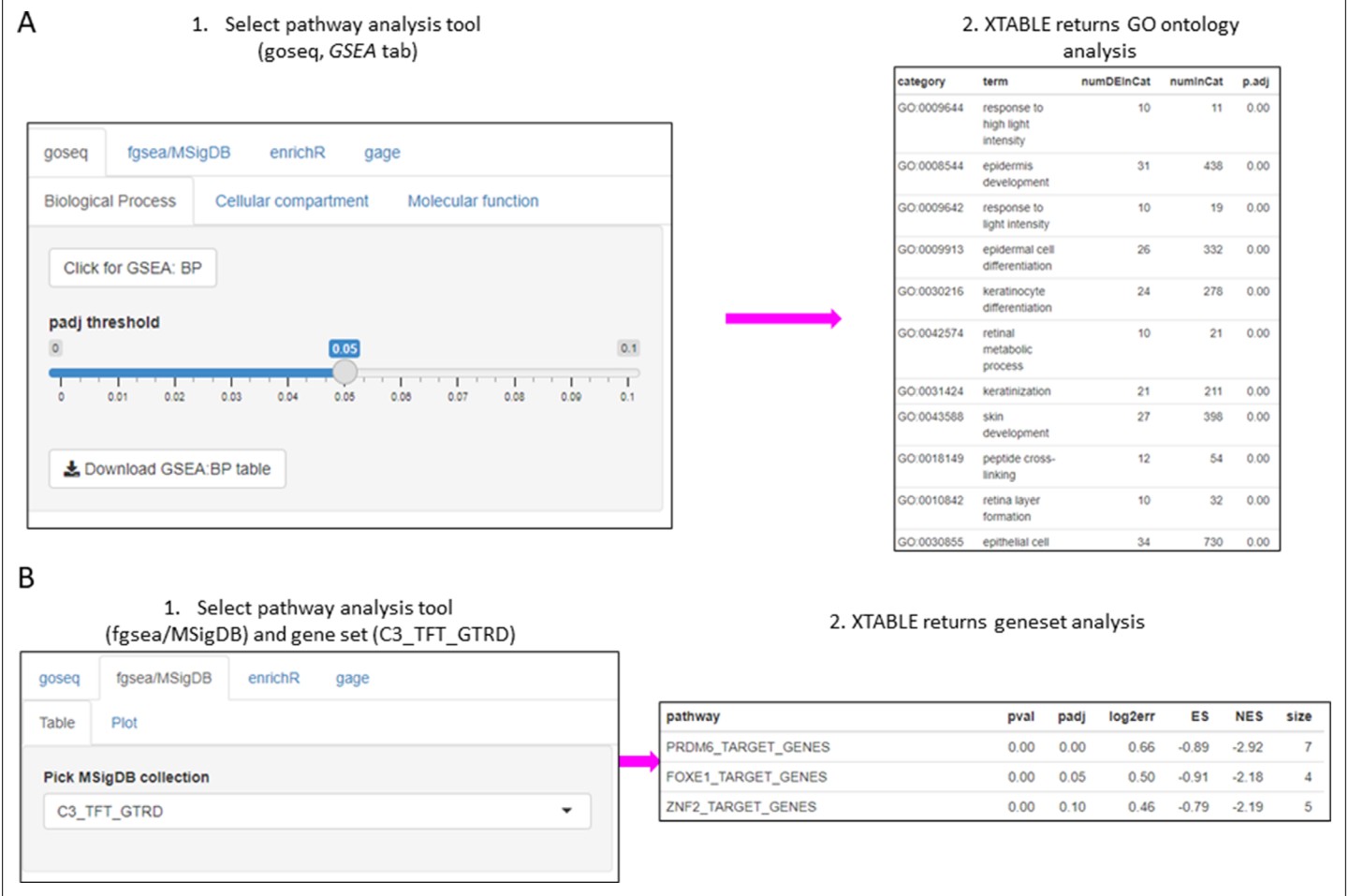

**Figure 4.** Gene set enrichment analyses in a list differentially expressed genes using the *GSEA* tab. (**A**) Gene set enrichment analysis using the goseq tool of a list of differentially expressed genes obtained in the *DEG* tab. One of the three main Gene Ontologies (GO) can be selected for analysis at a time. After selection of a p-value, XTABLE (**E**xploring **T**ranscriptomes of **B**ronchial **L**esions) returns a downloadable list of GO with associated statistics. (**B**) Gene set enrichment analysis using the fgsea/MSigDB tool. This tool allows the selection of any collection included in MSigDB and returns a list of signatures with associated statistics. The example shows the selection of the C3_TFT_GTRD collection (Transcription Factor Targets annotated in the Gene Transcription Regulation Database).

The online version of this article includes the following figure supplement(s) for figure 4:

**Figure supplement 1.** Example of pathway analysis (*PA* tab) output for a gene list obtained with the *DEG* function.

The list of differentially expressed genes obtained in the *DEG* tab can be automatically used in the *GSEA* and *PA* tabs to carry out gene set enrichment and pathway analyses. The *GSEA* tab allows the user to select the gene set enrichment tool (goseq, fgsea/MSigDB, enrichR, and gage), p-value cutoff, and the gene sets to analyse (*Figure 4A*). For instance, using the goseq function enables us to select one of the three Gene Ontology domains (biological process, cellular compartment, and molecular function) to consider for analysis (*Figure 4A*). Similarly, with the fgsea/MSigDB tool, the user can select the gene set of interest (*Figure 4B*). The *PA* tab operates in a similar manner using four pathway analysis tools (kegga/pathview, ReactomePA, Progeny, and Dorothea) (*Figure 4—figure supplement 1*).

## Gene-centred analysis and user-defined transcriptional signatures

Users can investigate their own gene or group of genes of interest in the application. To facilitate this type of gene-centred analyses, we have included the *Genes* and *Signature* functions in XTABLE.

In the *Genes* tab, the user can analyse the expression of one gene of interest. This tab is divided in three tools. The 'Expr' tool returns a downloadable list of the maximum (if multiple probes map to the

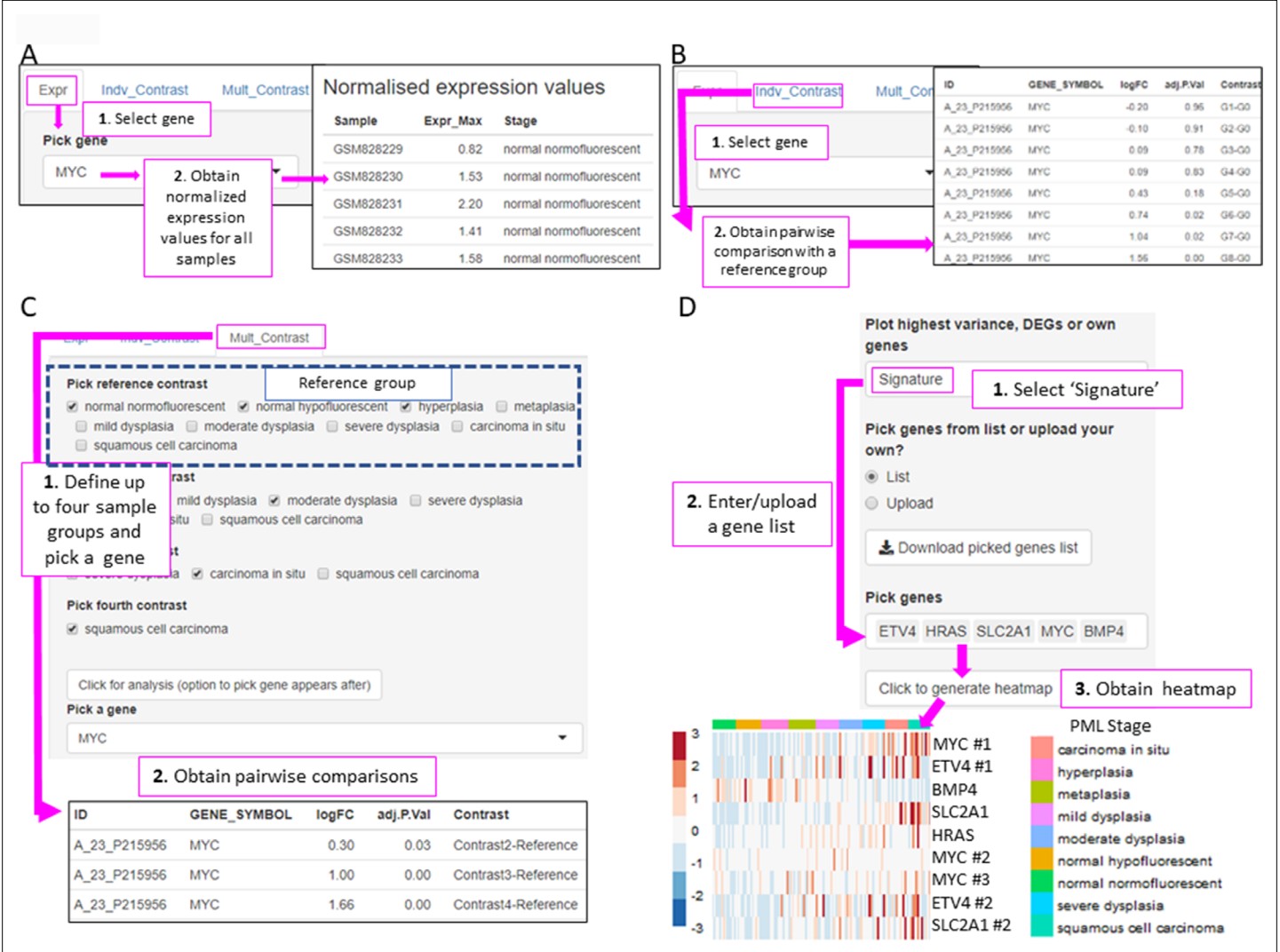

**Figure 5.** XTABLE (Exploring Transcriptomes of Bronchial Lesions) functions to implement analyses on individual genes (*Gene* tab) and user defined gene signatures (*Signature* tab). (**A**) The 'Expr' function (under the *Gene* tab) retrieves the normalized expression values for a gene of interest in all samples. (**B**) The 'Indv_Contrast' tool compares the expression of a gene of interest in groups of samples with a predetermined group. In the example, the function compares the expression of *MYC* in all stages with the normal normofluorescent group in GSE33479. (**C**) The 'Mult_Contrast' tool enables the grouping of samples in up to four groups (contrasts) and statistical comparison with a reference group determined by the user. The example shows the analysis of *MYC* expression in four groups of samples from the GSE33479 cohort (normal, low-grade, high-grade, and invasive carcinomas). The 'normal' group is set as reference group for statistical analysis. (**D**) Example of the use of the *Heatmap* tab to interrogate to visualize the expression of gene sets in premalignant lesions (PMLs). Gene sets can be defined by the user (as in the example) and are shown using the stage classification and entered manually or from a .csv file. Alternatively, the heatmap can be generated from a list of differentially expressed genes from the *DEG* tab or a selected number of genes filtered by variance. The three options can be selected in the scroll-down menu. In the example shown, the heatmap shows all microarray probes associated to each gene symbol. p-Values calculated using Welch's t-test.

same gene symbol) normalized expression values for the gene of interest in all samples (*Figure 5A*). The 'Indv_Contrast' function returns the differentially expressed gene analysis results including fold-change and statistical significance values for a given gene in all groups of samples compared with a predetermined reference group (*Figure 5B*). Sample grouping and reference group depend on the study. The 'Mult_Contrast' function works similarly to the 'Indv_Contrast' function but allows merging of up to four groups of samples and the reference group for statistical analysis can be determined by the user. The example in *Figure 5C* shows the evolution of MYC expression in four groups of samples that represent normal, low-grade PMLs, high-grade PMLs, and invasive carcinomas (*Figure 5C*). The fold-changes and p-values are the result with the comparison with the 'normal' group (normal normo-fluorescent, normal hypofluorescent, and hyperplasia) as the reference group (*Figure 5C*).

To facilitate the interrogation of biological processes driving PML progression, XTABLE also allows the interrogation of transcriptomic signatures using multiple functions. The *Signature* tab returns a list of normalized expression values and a graph with the signature scores (sum of log-normalized expression values) for a gene set determined by the user in all samples of the selected study.

The *Heatmap* tab returns a heatmap that displays the expression values in all samples in a selected study. The gene set shown in the heatmap can be selected using three options from the scroll-down menu. With the 'Signature' option, the user can manually enter or upload a list of genes. The example in *Figure 5D* shows a heatmap generated from the GSE33479 dataset with five transcriptional targets of *SOX2*, an important LUSC driver. The 'DEG' option automatically selects the list of genes differentially expressed in the *DEG* tab. Finally, the 'Variance' option selects genes using a user-defined number of genes with the highest variance.

## Assessment of the correlation between CIN signatures and progression potential

Two studies have highlighted the potential role of CIN as predictor of low-grade (*van Boerdonk et al., 2014*) and high-grade (*Teixeira et al., 2019*) PMLs progression in LUSC. XTABLE can be used to explore this correlation in the four cohorts, identify genes and pathways altered by CIN, and involved in driving it. Furthermore, XTABLE enables the user to carry out cross-comparisons between cohorts to identify high-confidence signals. One interesting example of this cross-comparisons is the correlation between CIN-scores and progression in different cohorts using the *CIN-score* and *ROC* tabs. Cohort GSE108124 focuses on CIS lesions with known progressive potential. As reported in that study (*Teixeira et al., 2019*), we found that the CIN5 signature segregates progressive and regressive lesions (AUC = 1) (*Figure 6A*, *Figure 6—figure supplement 1*), whereas CIN70 and CIN25 signatures are somewhat poorer predictors of progression (*Figure 6—figure supplement 2*) (AUC = 0.82 and 0.81, respectively). Apart from the varied performance of different CIN-scores, this difference can also be attributed to the lack of data about some of the genes in the sequencing output. Furthermore, cohort GSE114489, which also contains dysplastic samples with known progression potential, demonstrates that although signatures are elevated in persistent dysplasias, neither CIN5 (AUC = 0.72) (*Figure 6B*, *Figure 6—figure supplement 3*) nor CIN70 (AUC = 0.74) and CIN25 (AUC = 0.73) (*Figure 6—figure supplement 4*) could accurately segregate persistent from regressive samples as efficiently as CIN5 in the GSE108124 cohort. Similarly, CIN-scores did not accurately segregate regressive from persistent/progressive samples by progression status in the discovery cohort of GSE109743 (*Figure 6C*, *Figure 6—figure supplement 5*). A similar result was found with the validation cohort of GSE109743 (*Figure 6—figure supplement 6*). However, in cohorts with multiple stages represented (GSE109743 and GSE33479), there was a clear increase in CIN-scores with increasing PML grade (*Figure 6D and E*, *Figure 6—figure supplement 7*), which is consistent with the increased risk of malignant progression in high-grade lesions. In summary, we found that CIN-scores (specifically CIN5) are good predictors of malignant progression in CIS (GSE108124). In GSE114489, we observed an increase in the CIN5 score in persistent dysplasias but the discrimination between persistent and regressive dysplasias was poorer. In GSE109743, we did not observe significant differences between progressive/persistent PMLs and regressive lesions. This different performance of CIN-scores in different cohorts can be attributed to multiple factors both technical and biological. The most important differences are the differing definitions of progression status, and that GSE108124 focuses on CIS lesions, a high-grade precursor of invasive LUSC, whereas the other cohorts focus on earlier lesions (GSE114489) or combinations of different stages (GSE109743). Additionally, microarray analysis was carried out with RNA extracted from microdissected PMLs in cohort GSE108124 providing an enriched epithelial signal.

## Mapping the evolution of the most relevant pathways involved in LUSC using XTABLE

Inactivation of the tumour suppressor genes *TP53* and *CDKN2A* (*Figure 7A*) are the most frequent somatic events in LUSC (*Jamal-Hanjani et al., 2017*; *The Cancer Genome Atlas Research Network, 2012*). Other somatic alterations in driver genes are found at a lower frequency but often target the same pathways in different ways (*Figure 7A*; *Jamal-Hanjani et al., 2017*; *The Cancer Genome Atlas Research Network, 2012*). The squamous differentiation, PI3K/Akt, and oxidative stress response are the most frequently targeted pathways in LUSC (*The Cancer Genome Atlas Research Network,*

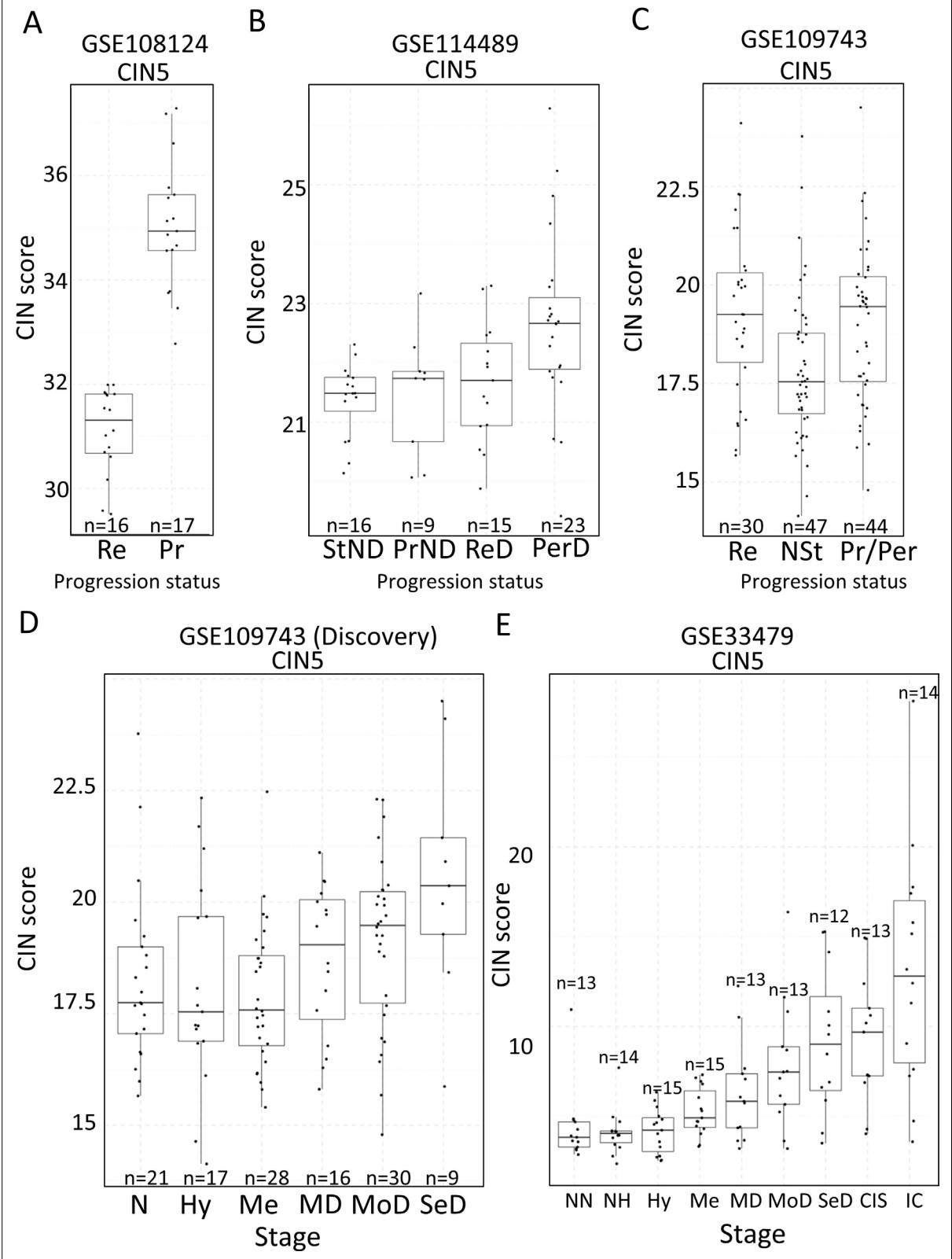

**Figure 6.** Association of carcinomas in situ (CIN)-scores with progression status and stage in the four cohorts of XTABLE (Exploring Transcriptomes of Bronchial Lesions). (**A**) CIN5 score in regressive (Re) and progressive (Pr) carcinomas in situ (CIS) lesions from cohort GSE108124. (**B**) CIN5 scores in stable non-dysplasias (StND), progressive non-dysplasias (PrND), regressive dysplasias (ReD), and persistent dysplasias (PerD) from cohort GSE114489. (**C**) CIN5 scores in Re, normal-stable (NSt), and progressive/persistent (Pr/Per) premalignant lesions (PMLs) from cohort GSE109743. (**D and E**) Evolution of CIN-

*Figure 6 continued on next page*

*Figure 6 continued*

scores in lung squamous cell carcinoma (LUSC) developmental stages for cohorts GSE109743 and GSE33479. N: normal; NN: normal normofluorescent; NH: normal hypofluorescent; Hy: hyperplasia; Me:metaplasia; MD: mild dysplasia; MoD: moderate dysplasia; SeD: severe dysplasia; CIS: carcinoma in situ; IC: invasive carcinoma. Boxplots show median and upper/lower quartile. Whiskers show the smallest and largest observations within 1.5× IQR.

The online version of this article includes the following figure supplement(s) for figure 6:

**Figure supplement 1.** Receiver operating characteristic (ROC) analysis of CIN5 as predictor of carcinomas in situ (CIS) progression in the GSE108124 cohort.

**Figure supplement 2.** Analysis of CIN70 and CIN25 scores as predictors of carcinomas in situ (CIS) progression in GSE108124.

**Figure supplement 3.** Receiver operating characteristic (ROC) analysis of CIN5 as predictor of premalignant lesion (PML) progression in the GSE114489 cohort.

**Figure supplement 4.** Analysis of CIN70 and CIN25 scores as predictors of premalignant lesion (PML) progression in GSE114489.

**Figure supplement 5.** Analysis of CIN70, CIN25, and CIN5 scores as predictors of premalignant lesion (PML) progression in GSE109743.

**Figure supplement 6.** CIN5 scores in the GSE109743 cohort with samples classified by progression status.

**Figure supplement 7.** Evolution of CIN5 scores by premalignant lesion (PML) stage in the validation cohort of GSE109743.

*2012*). The most frequent alteration targeting the squamous differentiation pathway is *SOX2* amplification, although inactivations of NOTCH proteins have also been proposed to target this pathway as they are mutually exclusive with *SOX2* amplification. A similar pattern of mutually exclusive somatic alterations targeting the same pathway has also been observed in the PI3K/Akt and oxidative stress response pathways (*Figure 7A*; *The Cancer Genome Atlas Research Network, 2012*). The role of these pathways in the transition between the LUSC developmental stages has not been addressed to date. The main reason for this is the paucity of genomic characterizations of PMLs and the lack of preclinical models of PMLs. Using XTABLE (*Signature* tab) to interrogate published transcriptional signatures correlated with these pathways can shed information about the stages at which they become active, and therefore what their potential role is in LUSC progression. To map changes in the activation of these three pathways to LUSC developmental stages, we use pre-designed transcriptional signatures from the MSigDB collections (*Subramanian et al., 2005*; *Liberzon et al., 2011*; *Liberzon et al., 2015*). Namely, the SOX2_BENPORATH (squamous differentiation) (*Ben-Porath et al., 2008*), HALLMARK_PI3K_AKT_MTOR_PATHWAY (PI3k/AKT pathway), and WP_NRF2_PATHWAY (oxidative stress response) signatures (*Figure 7B, C and D*, respectively). When GSE33479 was interrogated, the signature scores of the three pathways increased significantly in either moderate or severe dysplasias when compared with normal normofluorescent mucosa. No significant increases were detected in mild dysplasias or earlier lesions. Similar results were observed with the GSE109743 cohort except for the WP_NRF2_PATHWAY (*Figure 7—figure supplement 1*). In this pathway, an increase of the signature was detected only in mild dysplasias compared with normal samples, and no significant changes were detected in any other PMLs stages.

We also interrogated the MSigDB collection to investigate the stages wherein the activation of CDK4/cyclin-D1 activity is detected using an associated transcriptional signature. The CDK4/cyclin-D1 axis controls the transition through the G1-phase of the cell cycle and is frequently altered in LUSC by multiple mechanisms such as *CDKN2A* inactivation (which encodes the p16INK[4a] tumour suppressor and inhibits the activity of the CDK4/cyclin-D1 complexes and E2F transcriptional activity), and cyclin-D1 amplification. Activation of transcriptional signatures associated with these cell cycle regulators (MOLENAAR_TARGETS_OF_CCND1_AND_CDK4_DN and HALLMARK_E2F_TARGETS) can be used to monitor CDK4/cyclin-D1 dysregulation (*Molenaar et al., 2008*). The CDK4/cyclin-D1 signature showed a significant increase in moderate dysplasias and later stages in cohort GSE33479 (*Figure 7E*) and in mild dysplasias and later stages in the GSE109743 (*Figure 7—figure supplement 1*). The increase of E2F signature was already detectable in metaplasias and later stages (*Figure 7F*) in the GSE33479 cohort, whereas in the GSE109743 cohort the increase in the E2F signature was observed starting in mild dysplasias (*Figure 7—figure supplement 1*).

Overall, these results show that the activity of squamous differentiation, PI3K/Akt, and pathways start in the transition between high- and low-grade lesions (*Figure 1*), typically moderate and severe dysplasias, indicating a role of these pathways in this transition, whereas their function in earlier stages (e.g., transition from normal epithelium to low-grade PMLs) is likely to be more limited. Our

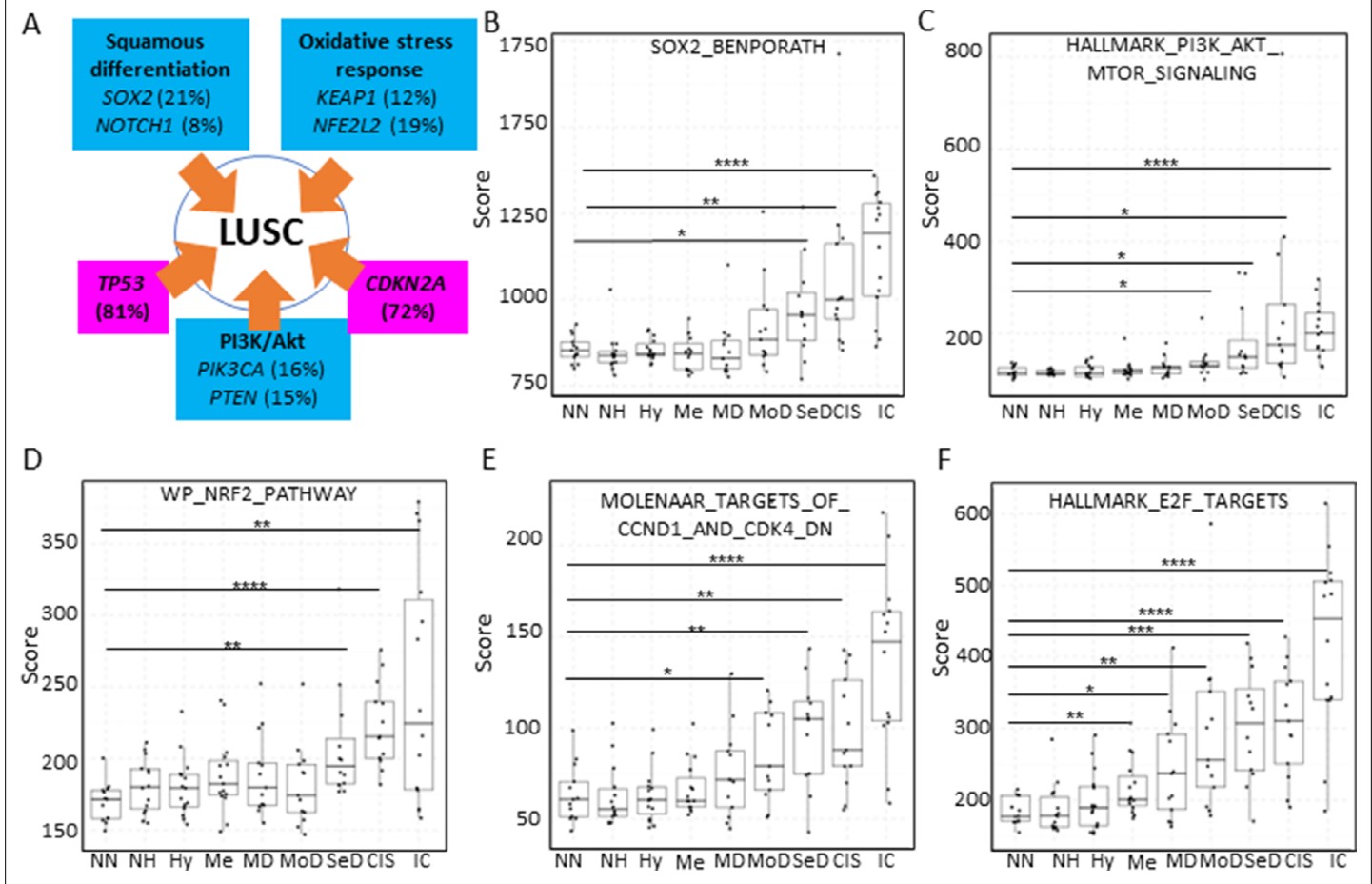

**Figure 7.** Mapping the evolution of the most relevant lung squamous cell carcinoma (LUSC) pathways to the LUSC developmental stages using published MSigDB transcriptional signatures. (**A**) Diagram showing the most important pathways involved in LUSC and the genes involved in such pathways that are found genetically altered in LUSC tumours. (**B**) Evolution of the SOX2 (the most frequent driver of the squamous differentiation pathway) transcriptional signature (SOX2_BENPORATH) during LUSC progression (GSE33479 cohort). (**C**) Evolution of the PI3K/Akt pathway during LUSC progression (HALLMARK_PI3K_AKT_MTOR_SIGNALING). (**D**) Evolution of the NRF2 (WP_NRF2_PATHWAY) transcriptional signature (correlated with the oxidative stress response) during LUSC progression. (**E**) Evolution of a transcriptional signature correlated with cyclin-D1 and CDK4 (MOLENAAR_TARGETS_OF_CCND1_AND_CDK4_DN) during LUSC progression. *CDKN2A* alterations in LUSC lead to the inactivation of the p16INK[4a], a CDK4 inhibitor. (**F**) Evolution of the expression of E2F targets (HALLMARK_E2F_TARGETS). Sample size: NN n=13, NH n=14, Hy n=15, Me n=15, MD n=13, MoD n=13, SeD n=12, CIS n=13, IC n=14. Boxplots show median and upper/lower quartile. Whiskers show the smallest and largest observations within 1.5× IQR. *p<0.05, **p<0.01, ***p<0.001, p<0.0001 (Welch's t-test).

The online version of this article includes the following figure supplement(s) for figure 7:

**Figure supplement 1.** Evolution of five transcriptional signatures in cohort GSE109743.

observations with the CDK4/cyclin-D1 and E2F signatures indicate that the onset of this pathways start in earlier stages (metaplasias and mild dysplasias).

## Discussion

In this report, we have described XTABLE, an open source bioinformatic tool to explore gene expression in LUSC PMLs using four different transcriptomic datasets. The most novel aspect of this application is the emphasis on preinvasive disease (to our knowledge, the first bioinformatic tool focusing on PMLs), and the possibility of multiple sample stratifications by parameters that correlate with high risk of malignant progression (stage, progression status, and CIN). Treating lung cancer is complex. Chemotherapies, radiotherapy, targeted therapies, and immunotherapies save lives. However, the progress in lung cancer patient survival during the last 20 years is disappointing (*National Cancer Institute and DCCPS, 2018*). Redirecting research efforts to prevent lung cancer and to detect its more treatable premalignant stages

is the most efficient way to prevent lung cancer deaths to date. XTABLE offers researchers the possibility of accessing the most relevant transcriptomic databases on LUSC PMLs to assist in the understanding of PML biology and identify biomarkers for LUSC early detection. Biomarker identification, investigating the evolution of signalling pathways in multiple developmental LUSC stages, identification of immunomodulatory signals, changes in transcriptional signatures, and exploring the causes and consequences of CIN in PMLs are amongst the multiple examples of promising applications of XTABLE for basic and translational biologists.

Whereas multiple open-source applications for generic processing of transcriptomic data have been developed since the advent of microarray and RNAseq technologies, none of those applications integrate analysis capabilities directed to prevention and early detection discovery research. Furthermore, these applications lack versatility for downstream analysis, automatically apply data transformation (log-transformation) to dataset that do not require it, lack of update for years, compatibility issues with current operating systems, require reformatting and renaming of expression datasets (not practical for large sample numbers), and do not evaluate the diagnostic performance of classifiers such as ROC analysis. XTABLE overcomes these limitations and unifies different packages in a manner that facilitates precancerous biology, prevention, and early detection research.

XTABLE also contributes to a more open, accessible, and inclusive science. Research laboratories often have restricted access to bioinformatic support due to funding constraints or lack of adequate collaborations. This limitation can be a major hurdle in the competitiveness of research groups as it may prevent hypothesis generation, validation of experimental results in patient cohorts, or acquisition of preliminary results. XTABLE and similar applications can contribute to addressing those disadvantages with an accessible and versatile platform for gene expression analysis. This application also contributes to the open science philosophy as it promotes the dissemination of data, accessibility, and transparency. Although XTABLE is unlikely to offer all the possibilities of analysis required by the scientific community, the code can be obtained by the users to adapt it to their research questions. Additionally, XTABLE allows the download of results that can be subject to additional downstream analyses.

Although integrative analyses of two or more datasets as well as integration of other genomic platforms would be desirable capabilities of XTABLE, the different nature of the studies, methodologies, and platforms make these integrations hardly achievable. For instance, stratifications that apply to all datasets and setting up the same thresholds for ROC analysis is not possible. Moreover, implementing separate analyses is the most scientifically sound alternative to prevent wrong conclusions. Nevertheless, by analysing the four datasets under a unified interface, and at the same time, implementing analyses on individual datasets, XTABLE achieves an optimal balance between clarity, speed of analysis, user friendliness, and scientific rigor while avoiding overcomplications. Additionally, only one study (GSE108105) also published genomic and epigenomic data.

Another potential limitation of XTABLE is that it cannot be used to interrogate future datasets in a simple manner for non-computational scientists. This limitation arises from the fact that the characteristics of future studies are hardly predictable, which hampers the design of stratification and analysis options. However, the open-source nature of this application allows the future modification of the code to add new datasets by other users. Moreover, the analysis modules for each of the current databases provide a template that will be re-coded to integrate future databases.

Our analysis of CIN signatures and their correlation with the known progression potential of PMLs is one of the examples of the use of XTABLE to obtain a wider view of LUSC PML biology. CIN5 was a good predictor of CIS progression in the GSE108124 study, as already described by the authors of the study (*Teixeira et al., 2019*). However, CIN5 did not perform as well in other studies. Conceivably, the most plausible reason for this discrepancy is the different definition of progression status in the four studies. Whereas GSE108124 provides a binary classification of PMLs (progressive and regressive), the other studies show a more complex classification that includes progressive and persistent lesions under the same category. The endpoint in the definition of the progression status also differs between studies. Specifically, the endpoint in GSE108124 is progression to invasive LUSC whereas the other studies define progression as transition to a higher grade. Conceivably, CIN might not play the same role in the transition between PMLs as it does in malignant transformation. Additionally, GSE108124 focuses on CIS, the precursor lesions of invasive carcinomas, and uses microdissected samples. The other two studies with progression status information focus on different stages and do not perform enrichment of the tumour component. GSE114489 investigates dysplasias, an earlier LUSC stage and

lacks further subdivision into mild, moderate, and severe dysplasic lesions. This limitation might result in data with more noise and poorer correlation with progression status. Nevertheless, this cohort did show an increase in CIN5 signature in persistent dysplasias when compared with regressive lesions, a difference that is not observed in GSE109743 (a cohort that does not separate PMLs by stage). These discrepancies point at several limitations of the use of CIN and its associated signatures as surrogates of progression potential. The use of CIN and more specifically CIN-scores as a bona fide predictor of progression might be limited to microdissected samples and CIS lesions. For example, the presence of tumour stroma could result in an underestimated CIN-score. Additionally, CIS lesions are the most advanced premalignant stage and show the highest levels of CIN5 signatures, thereby contributing to a higher CIN-related signal. Comparisons with microdissected PMLs of earlier stages are necessary to address the extent of these limitations.

Regardless, CIN gene expression signatures present a very robust correlation with genomic instability indexes in cancer, and specifically in lung cancer (*Carter et al., 2006*). Although we have shown that the correlation of CIN signatures with progression potential varies between databases, they can be used to investigate other biological questions such as identification of CIN-tolerance and CIN-driver genes, identification of changes in the immune microenvironment associated with CIN (in light of the new role of CIN in modulating tumour immunity *Tijhuis et al., 2019*), and interrogation of genes of interest for the user in CIN-high vs. CIN-low PMLs.

Using XTABLE, we have mapped the most important signalling pathways targeted in LUSC to the developmental stages. We found that the onset activation of squamous differentiation, PI3K/Akt pathways occur in the transition from low- to high-grade PMLs. Comprehensive genomic characterizations of PMLs have not been undertaken so far, and therefore, we are not able to map the onset of LUSC pathways with the genomic profiles of each premalignant stage. However, our observations suggest that the genomic alterations targeting those pathways should be more frequent in high-grade PMLs than low-grade. Another explanation is that those mutations are present in low-grade PMLs but their effect on the pathways is only unleashed in the transition to high-grade lesions. Co-occurring somatic alterations and microenvironment changes could explain this. Nevertheless, our results strongly indicate that squamous differentiation and PI3K/Akt pathways are unlikely to play a role in the earliest developmental stages (hyperplasias, metaplasias, and mild dysplasias). On the other hand, our observations with the CDK4/cyclin-D1 and E2F signatures indicate an earlier onset (metaplasias and moderate dysplasias). Both signatures can be influenced by *CDKN2A* inactivation and cyclin-D1 (*CCND1*) amplification, two alterations frequently observed in LUSC, and known to override oncogene-induced senescence (*Serrano et al., 1997*). It is conceivable that *CDKN2A* inactivation and *CCND1* amplification occur earlier than activation of oncogenic in order to prevent oncogene-induced senescence. For instance, *SOX2* overexpression has a negative effect in cell fitness in multiple experimental scenarios (*Cho et al., 2013*; *Correia et al., 2017*; *Cox et al., 2012*; *Wuebben et al., 2016*). Early activation of the CDK4/cyclin-D1 pathway could be key to avoid this toxicity in LUSC. A similar scenario is also likely to occur with PI3K/Akt signalling, as this pathway needs to be exquisitely regulated to avoid senescence (*Astle et al., 2012*). However, mechanisms that modify CDK4/cyclin-D1 and E2F activities independently of *CDKN2A* inactivation cannot be ruled out as multiple oncogenic pathways can regulate the cell cycle. Furthermore, the existence of *CDKN2A* inactivation in low-grade PML is not known.

The activation of the squamous differentiation, PI3K/Akt pathways in high-grade PMLs could be very useful as biomarkers to develop new modalities of early detection using multiple approaches. High-grade PMLs frequently undergo malignant progression (*Ishizumi et al., 2010*) and therefore, detection of these lesions and removal is an optimal strategy to prevent deaths by LUSC. Design of appropriate molecular probes and theragnostic technologies that identify lesions with high-level pathway activation, and secreted proteins that are associated with those pathways are amongst the strategies to exploit pathway activation in early detection.

Finally, the aim of this article is not to prioritize any of the studies included in XTABLE, but to provide a tool for the simple and quick analysis of a large amount of biologically relevant data on PML biology. Each study has its own advantages and limitations. Therefore, the user is ultimately responsible for choosing the datasets that are more adequate to interrogate their research questions, compare results between databases considering the different scientific contexts of each study, and interpret them in light of the different designs of each study.

## Materials and methods

### XTABLE download and installation

XTABLE can be downloaded from the GitLab repository (https://gitlab.com/cruk-mi/XTABLE) and it requires the previous installation of RStudio. Copy the commands, paste them on the RStudio console, and run the command (*Video 1*).

### XTABLE packages and construction

Bioconductor package GEOquery (2.54.1) was used to retrieve the data and the Bioconductor package Biobase (2.46.0) used to extract the gene expression values for microarray datasets. The Bioconductor package limma (3.42.2) was used to generate differentially expressed gene analysis results. Bioconductor packages AnnotationDbi (1.48.0) and org.Hs.eg.db (3.10.0) were used to retrieve additional gene IDs. Additional gene IDs were retrieved from Ensembl BioMart website (https://www.ensembl.org/biomart/martview/) with Ensembl Genes 104 dataset and Human Genes GRCh38.p13 to generate four mapping files with the following attributes: 'Gene stable ID' and 'AGILENT WholeGenome 4×44k v1 probe'; 'Gene stable ID' and 'Transcript stable ID'; 'Gene stable ID' and 'Gene name'; 'Gene stable ID' and 'NCBI gene (formerly Entrezgene) ID' for GSE33479. For GSE114489, a file containing the attributes 'Gene stable ID' and 'AFFY HuGene 1 0 st v1 probe' was downloaded. The Bioconductor package edgeR (3.28.1) was used to calculate CPM values. R package pROC (1.17.0.1) was used to calculate AUC and generate ROC curves. Gene set enrichment analysis and pathway analysis were performed using Bioconductor packages ideal (1.10.0), fgsea (1.14) with MSigDB (https://www.gsea-msigdb.org/gsea/msigdb/collections.jsp) gene set collections (7.1), limma (3.42.2), pathview (1.26.0), enrichR (3.0), gage (2.36.0) with gageData (2.24.0), ReactomePA (1.30), progeny (1.8.0), and dorothea (0.99.0).

Deconvolution analysis was performed for microarray data using estimate (1.0.11) and for RNAseq data using imsig (1.1.3). imsig requires filtering out genes with low variance which was performed with the package matrixStats (0.59.0). R package stats (3.6.0) used to generate PCA data. ggplot2 (3.3.3), pheatmap (1.0.12), RColorBrewer (1.1–2) were used for making plots and ggpubr (0.4.0) was used to perform Welch's t-test. tidyr (1.1.3), tibble (3.1.2), dplyr (2.0.6), and magrittr (2.0.1) were used for general data processing and formatting. Code was written using RStudio Workbench (1.4.1717.3) using R (4.0.3).

### Transcriptional signatures

To investigate the evolution of LUSC pathways, we downloaded the following gene sets from the Molecular Signatures Database (MSigDB v7.5.1):

BENPORATH_SOX2_TARGETS (https://www.gsea-msigdb.org/gsea/msigdb/cards/BENPORATH_SOX2_TARGETS.html).

HALLMARK_PI3K_AKT_MTOR_SIGNALING (https://www.gsea-msigdb.org/gsea/msigdb/cards/HALLMARK_PI3K_AKT_MTOR_SIGNALING.html).

WP_NRF2_PATHWAY (https://www.gsea-msigdb.org/gsea/msigdb/cards/WP_NRF2_PATHWAY.html).

MOLENAAR_TARGETS_OF_CCND1_AND_CDK4_DN (https://www.gsea-msigdb.org/gsea/msigdb/cards/MOLENAAR_TARGETS_OF_CCND1_AND_CDK4_DN.html).

HALLMARK_E2F_TARGETS (https://www.gsea-msigdb.org/gsea/msigdb/cards/HALLMARK_E2F_TARGETS.html).

The signatures were manually curated to replace unrecognized gene symbols with alternative symbols used in each cohort. The signature scores returned by the XTABLE were calculated as the sum of log-normalized expression values in each cohort.

## Acknowledgements

This work has been carried out with funding provided by the Cancer Research UK Lung Cancer Centre of Excellence (A25146) and the Cancer Research UK Manchester Institute (A27412) and the Manchester Biomedical Research Centre.

We would like to thank the Scientific Computing (SciCom) team of the CRUK Manchester Institute for their support in this project.

Representative images of premalignant stages in *Figure 1* are shown with permission of the Manchester Cancer Research Centre (MCRC) Biobank, UK. The MCRC Biobank holds a generic ethics approval (Ref: 18/NW/0092) which can confer this approval to users of banked samples via the MCRC Biobank Access Policy.

## Additional information

### Funding

| Funder | Grant reference number | Author |
|---|---|---|
| Cancer Research UK | A25146 | Julia Ogden<br>AS Mukarram Hossain<br>Anshuman Chaturvedi<br>Alastair RW Kerr<br>Caroline Dive |
| Manchester Biomedical Research Centre | | Matthew Roberts |

The funders had no role in study design, data collection and interpretation, or the decision to submit the work for publication.

### Author contributions

Matthew Roberts, Conceptualization, Resources, Data curation, Software, Formal analysis, Validation, Investigation, Visualization, Methodology, Writing - original draft, Writing – review and editing; Julia Ogden, Conceptualization, Validation, Investigation, Writing – review and editing; AS Mukarram Hossain, Software, Validation, Investigation, Methodology; Anshuman Chaturvedi, Data curation, Formal analysis, Validation, Writing – review and editing; Alastair RW Kerr, Software, Supervision, Validation, Investigation, Methodology, Writing – review and editing; Caroline Dive, Supervision, Funding acquisition, Investigation; Jennifer Ellen Beane, Conceptualization, Validation, Investigation, Methodology, Writing – review and editing; Carlos Lopez-Garcia, Conceptualization, Resources, Data curation, Software, Formal analysis, Supervision, Validation, Investigation, Visualization, Methodology, Writing - original draft, Project administration, Writing – review and editing

### Author ORCIDs

Alastair RW Kerr http://orcid.org/0000-0001-9207-6050
Carlos Lopez-Garcia http://orcid.org/0000-0001-9848-8216

### Decision letter and Author response

Decision letter https://doi.org/10.7554/eLife.77507.sa1
Author response https://doi.org/10.7554/eLife.77507.sa2

## Additional files

### Supplementary files
• MDAR checklist

### Data availability
The current manuscript makes use of previously published databases, so no data have been generated for this manuscript. All analyses shown in the manuscript has been carried out using XTABLE and can be reproduced easily by any user.

The following previously published datasets were used:

| Author(s) | Year | Dataset title | Dataset URL | Database and Identifier |
|---|---|---|---|---|
| Mascaux C | 2014 | Molecular characterisation of the multistep process of lung squamous carcinogenesis by gene expression profiling | https://www.ncbi.nlm.nih.gov/geo/query/acc.cgi?acc=GSE33479 | NCBI Gene Expression Omnibus, GSE33479 |
| Beane JE, Spira A | 2019 | Bronchial premalignant lesions have distinct molecular subtypes associated with future histologic progression | https://www.ncbi.nlm.nih.gov/geo/query/acc.cgi?acc=GSE109743 | NCBI Gene Expression Omnibus, GSE109743 |
| Merrick D | 2018 | Altered Cell-Cycle Control, Inflammation and Adhesion in High-Risk Persistent Bronchial Dysplasia | https://www.ncbi.nlm.nih.gov/geo/query/acc.cgi?acc=GSE114489 | NCBI Gene Expression Omnibus, GSE114489 |
| Teixeira VH | 2019 | Deciphering the genomic, epigenomic and transcriptomic landscapes of pre-invasive lung cancer lesions to determine prognosis | https://www.ncbi.nlm.nih.gov/geo/query/acc.cgi?acc=GSE108124 | NCBI Gene Expression Omnibus, GSE108124 |

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
