## [Editor Report]

The authors have created a resource tool that is valuable in assessing precancerous lesions in the lung, which may serve as a tool for investigators working in this area, and as an example for additional similar resources. The accessibility of the tool is a concern but does not diminish the quality of the product.

---

## [Decision Letter]

**Decision letter after peer review:**

[Editors’ note: the authors submitted for reconsideration following the decision after peer review. What follows is the decision letter after the first round of review.]

Thank you for submitting the paper "Interrogating the Precancerous Evolution of Pathway Dysfunction in Lung Squamous Cell Carcinoma Using XTABLE" for consideration by *eLife*. Your article has been reviewed by 2 peer reviewers, and the evaluation has been overseen by a Reviewing Editor and a Senior Editor.

We are sorry to say that, after consultation with the reviewers, we have decided that this work will not be considered further for publication by *eLife*. There was different opinion among the reviewers, and we sought a third outside opinion to make a final decision. The principal reason is the lack of novelty in this work, the lack of possibility of integration of other datasets of precancerous lesions, and the existence of many other similar applications that are user-friendly. Curation of an early cancer database across multiple cancer types, which provides a user-friendly application to perform integrative analyses across datasets, we suspect would be beyond the scope of this study, and would be necessary to have the user-friendliness of this application carry the novelty. We have included reviewers' comments.

*Reviewer #2 (Recommendations for the authors):*

Some of the referencing needs careful proofreading. We notice for example that reference 25, Teixeira et al., is labelled as 2019 in the bibliography, 2020 in the text (line 86), and 2018 in Figure 1.

Line 153 – this study used whole genome, not whole exome sequencing.

Figure 2A appears to be an old screenshot as the app is titled "NSCLC" rather than "XTABLE".

---

## [Author Response]

[Editors’ note: The authors appealed the original decision. What follows is the authors’ response to the first round of review.]

Reviewer #2 (Recommendations for the authors):Some of the referencing needs careful proofreading. We notice for example that reference 25, Teixeira et al., is labelled as 2019 in the bibliography, 2020 in the text (line 86), and 2018 in Figure 1.Line 153 – this study used whole genome, not whole exome sequencing.Figure 2A appears to be an old screenshot as the app is titled "NSCLC" rather than "XTABLE".

We will address these points accordingly.

All these points have been sorted out according to the reviewer’s comments.